# Quantitative analysis of mammalian GIRK2 channel regulation by G proteins, the signaling lipid PIP$_2$ and Na$^+$ in a reconstituted system

Weiwei Wang, Matthew R Whorton, Roderick MacKinnon*

Laboratory of Molecular Neurobiology and Biophysics, Howard Hughes Medical Institute, The Rockefeller University, New York, United States

**Abstract** GIRK channels control spike frequency in atrial pacemaker cells and inhibitory potentials in neurons. By directly responding to G proteins, PIP$_2$ and Na$^+$, GIRK is under the control of multiple signaling pathways. In this study, the mammalian GIRK2 channel has been purified and reconstituted in planar lipid membranes and effects of Gα, Gβγ, PIP$_2$ and Na$^+$ analyzed. Gβγ and PIP$_2$ must be present simultaneously to activate GIRK2. Na$^+$ is not essential but modulates the effect of Gβγ and PIP$_2$ over physiological concentrations. Gα$_{i1}$(GTPγS) has no effect, whereas Gα$_{i1}$(GDP) closes the channel through removal of Gβγ. In the presence of Gβγ, GIRK2 opens as a function of PIP$_2$ mole fraction with Hill coefficient 2.5 and an affinity that poises GIRK2 to respond to natural variations of PIP$_2$ concentration. The dual requirement for Gβγ and PIP$_2$ can help to explain why GIRK2 is activated by G$_{i/o}$, but not G$_q$ coupled GPCRs.

## Introduction

G protein gated inward rectifier K$^+$ (GIRK) channels are inward rectifier K$^+$ (Kir) channels whose activity is regulated by GTP binding proteins (G proteins) (*Brown and Birnbaumer, 1988*; *Yamada et al., 1998*). They are present in many different cell types including electrically excitable cells in the cardiovascular and nervous systems (*Dascal et al., 1993*; *Lesage et al., 1994*; *Krapivinsky et al., 1995*; *Lesage et al., 1995*; *Karschin et al., 1996*). GIRK channels regulate cellular electrical activity by modulating K$^+$ conductance near the resting membrane potential. Specifically, stimulation of certain G protein coupled receptors (GPCRs) on a cell's surface leads to the opening of GIRK channels, which increases K$^+$ conductance and opposes the initiation of action potentials. This inhibitory effect on excitation is best understood in the cardiovascular system, where vagus nerve stimulation slows heart rate through acetylcholine action on GPCRs in atrial pacemaker cells (*Loewi and Navratil, 1926*; *Giles and Noble, 1976*; *Osterrieder et al., 1981*). The ensuing activation of GIRK channels prolongs the interval between pacemaker action potentials, thus causing the heart rate to slow. GIRK channels are also abundant in the nervous system, where they modulate slow inhibitory postsynaptic currents in response to neurotransmitter stimulation (*Luscher et al., 1997*; *Lujan et al., 2009*).

Cellular electrophysiology experiments and biochemical analysis have yielded a rich understanding of GIRK channel function. As background to the questions addressed in the present study, the following conclusions have been established over the past several decades of study. First, the G protein 'subunit' called Gβγ (a tightly bound complex of β and γ protein polypeptides) is the major mediator through which GPCR stimulation opens GIRK channels (*Pfaffinger et al., 1985*; *Logothetis et al., 1987*; *Reuveny et al., 1994*; *Wickman et al., 1994*; *Huang et al., 1995*; *Inanobe et al., 1995*; *Kofuji et al., 1995*; *Krapivinsky et al., 1995*). When a GPCR is stimulated through the binding of a neurotransmitter (or drug) on the outside of the cell membrane it catalyzes guanine nucleotide exchange–GTP replaces

*For correspondence: mackinn@rockefeller.edu

**Competing interests:** The authors declare that no competing interests exist.

**eLife digest** Though every cell in the body is surrounded by a membrane, there are a number of ways that molecules can pass through this membrane to either enter or leave the cell. Proteins from the GIRK family form channels in the membranes of mammalian cells, and when open these channels allow potassium ions to flow through the membrane to control the membrane's voltage.

GIRK channels are found in the heart and in the central nervous system, and can be activated in a variety of ways. Sodium ions and molecules called 'signaling lipids' can regulate the activation of GIRK channels. These channels can also be caused to open by G proteins: proteins that are found inside cells and that help to transmit signals from the outside of a cell to the inside. Three G proteins—called Gα, Gβ, and Gγ—work together in a complex that functions a bit like a switch. When switched on, the Gα subunit is separated from the other two subunits (called Gβγ); and both parts can then activate different signaling pathways inside the cell.

The Gβγ subunits and a signaling lipid have been known to regulate the opening of GIRK channels for a number of years, but these events have only been studied in the context of living cells. The specific role of each molecule, and whether the Gα subunit can also regulate the GIRK channels, remains unknown. Now Wang et al. have produced one type of mouse GIRK channel, called GIRK2, in yeast cells, purified this protein, and added it into an artificial membrane. This 'reconstituted system' allowed the regulation of a GIRK channel to be investigated under more controlled conditions than in previous experiments.

Wang et al. found that the Gβγ subunits and the signaling lipid both need to be present to activate the GIRK2 channel. Sodium ions were not essential, but promoted further opening when Gβγ and the signaling lipid were already present. When locked in its 'on' state, the Gα subunit had no effect on GIRK2, but adding Gα locked in the 'off' state closed these channels by removing the Gβγ proteins.

The findings of Wang et al. suggest that it should be possible to use a similar reconstituted system to investigate what allows different G proteins to activate specific signaling pathways.

GDP on the Gα subunit–and release of Gα(GTP) and Gβγ subunits on the cytoplasmic side (*Gilman, 1987*). These subunits are then free to activate their 'target' proteins: Gβγ activates GIRK. Second, the signaling lipid phosphatidylinositol 4,5-bisphosphate ($PIP_2$) also regulates GIRK channel opening (*Huang et al., 1998*; *Sui et al., 1998*; *Logothetis and Zhang, 1999*). This property means that GIRK channels respond to more than one ligand and thus are under the control of more than one distinct signaling pathway. Third, the activity of certain GIRK channels, for example the neuronal GIRK2 channel, is also regulated by the concentration of intracellular $Na^+$ (*Petit-Jacques et al., 1999*; *Ho and Murrell-Lagnado, 1999a*, *1999b*). Intracellular $Na^+$ concentration increases during excessive electrical excitation. Activation of GIRK2 under this circumstance may serve as a cell protective mechanism through action potential suppression. Finally, x-ray crystallography has recently provided atomic level descriptions of a neuronal GIRK channel, GIRK2, in the absence and presence of Gβγ, $PIP_2$ and $Na^+$ (*Whorton and MacKinnon, 2011*, *2013*).

The present study addresses the following still unanswered questions. First, does the Gα(GTP) sub-unit also regulate GIRK function? The Gα subunit co-precipitates with GIRK in biochemical 'pull-down' experiments (*Huang et al., 1995*; *Peleg et al., 2002*; *Ivanina et al., 2004*; *Rubinstein et al., 2009*; *Berlin et al., 2010*, *2011*) and also exhibits interaction with the channel's cytoplasmic domain as detected by solution NMR spectroscopy (*Mase et al., 2012*). Do these interactions underlie channel regulation by Gα? Second, how does GIRK channel activity depend on the concentration of $PIP_2$ in the membrane? $PIP_2$ is a signaling lipid whose concentration in the cell membrane changes (*Lemmon, 2008*; *Balla, 2013*). To know whether GIRK activity is responsive to these changes, a quantitative description of the channel's $PIP_2$ concentration dependence is needed. Until now, no method has existed to change $PIP_2$ in a known, quantitative and controlled manner in membranes in which GIRK channel activity is measured. Third, do Gβγ, $PIP_2$ and $Na^+$, act on the channel independently or do they function together? The issue of multi-ligand dependence is relevant to understanding the molecular biophysics of channel regulation and to understanding how multiple signaling pathways control membrane potential. At the level of molecular biophysics we wish to understand how multiple allosteric inputs regulate channel

opening. At the level of cellular control of membrane potential we wish to understand GIRK's AND/OR relationship with respect to multiple signaling inputs.

The real difficulty in answering the above questions lies in the compositional complexity and uncertainty of living cell membranes—the natural system in which nearly all past experiments have been carried out. In this study we use the planar lipid bilayer technique to characterize the regulation of GIRK channels by G-proteins, PIP$_2$ and Na$^+$ ions. The bilayer technique is a total reconstitution method in which the channel and its various regulatory components are first purified to homogeneity and then combined. Every component of the system is known to within a small error in composition and amount, including the membrane, which is produced from either purified or synthetic lipids.

## Results

### Robust GIRK2 activity depends on the presence of both Gβγ and PIP$_2$ in the membrane

A schematic of the bilayer system depicts two chambers separated by a partition with a hole onto which lipid membranes are painted (*Montal and Mueller, 1972*; *Miller and Racker, 1976*; *Figure 1B*). Components are added by fusing lipid vesicles that contain channels and membrane-anchored G proteins and by adding Na$^+$ or soluble short chain PIP$_2$ molecules to either chamber. The chambers are connected electrically through a voltage-clamp circuit to record K$^+$ currents across the membrane.

The current trace in *Figure 2A* shows K$^+$ currents in bilayer membranes in which GIRK2 channel-containing vesicles were fused in the absence of PIP$_2$ and Gβγ. Brief channel openings appear as downward current spikes because the membrane voltage was held positive relative to ground (top chamber, *Figure 1B*) and the sign of current was inverted to conform to eletrophysiological convention. When PIP$_2$ was added to the top chamber half way through the trace, channel activity remained essentially unchanged (*Figure 2B*). Similarly, addition of Gβγ in the absence of PIP$_2$ did not perceptibly alter baseline channel activity (*Figure 2C*). Only when both PIP$_2$ and Gβγ were present, independent of the order in which they were added, did we observe large K$^+$ currents due to robust GIRK2 channel opening (*Figure 2D,E*). We note that the baseline current prior to the addition of the second ligand in *Figure 2D,E*, shown on an expanded current axis scale, is similar in magnitude to the currents in

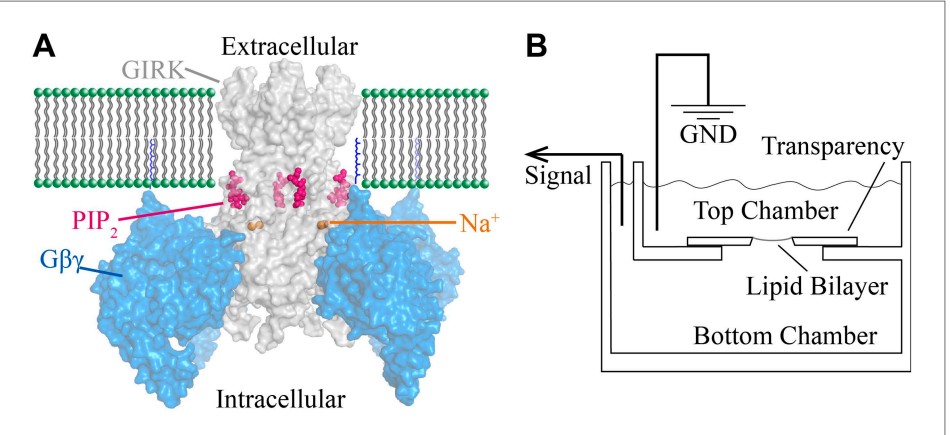

**Figure 1**. Structural depiction of GIRK2 regulation and horizontal planar bilayer configuration. (**A**) View from the membrane plane of a GIRK2 channel (gray) bound to its activating ligands PIP$_2$ (purple), Na$^+$ (brown) and the heterodimeric G protein subunit Gβγ (blue, with the postulated geranylgeranyl group on the Gγ subunit drawn as blue lines) in surface representation (PDBID: 4KFM). There are four binding sites for each of these ligands on each GIRK2 homotetramer. (**B**) A schematic of the horizontal planar bilayer system used to characterize GIRK2 channel activity. Two solution-filled chambers in a polyoxymethylene block are separated by a piece of transparency film with a small hole (~100 μm diameter). Lipid bilayers are formed across the hole by spontaneous thinning of a painted solution of lipid in decane. Membrane proteins are incorporated into the lipid bilayer by fusion of applied proteoliposomes. Soluble reagents are either directly applied to either chamber followed by thorough mixing or added via a local perfusion system. The two chambers are electrically voltage-clamped as indicated for current recordings.

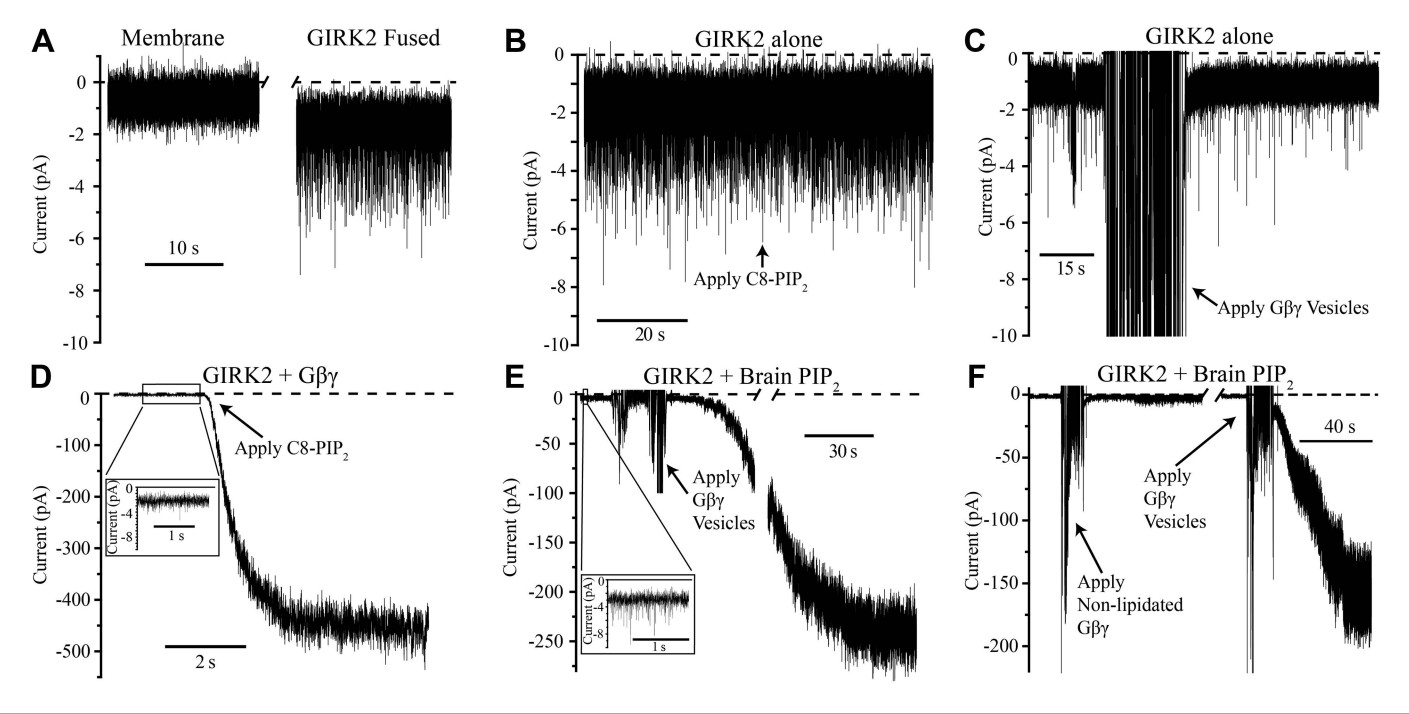

**Figure 2**. GIRK2 activation requires both Gβγ and PIP$_2$. Currents are plotted according to electrophysiology convention such that negative values represent inward current with respect to channel orientation. The same buffer containing 150 mM KCl was used in both chambers and the membranes were held at −50 mV (eliciting inward channel current). Initial compositions of each bilayer are indicated above each panel. Fusion of lipid vesicles was done manually and induced significant noise during application. This noise did not interfere with subsequent recording of current change. (**A**) Current spikes were observed after fusing GIRK2 vesicles into the lipid bilayer. (**B** and **D**) Gβγ is necessary for PIP$_2$ activation of GIRK2. Currents were recorded from a bilayer containing either GIRK2 alone (**B**) or GIRK2+Gβγ (**D**) before and after application of 32 μM C8-PIP$_2$ (indicated with arrow). Only when Gβγ was present (**D**) did application of PIP$_2$ increase current through GIRK2. (**C** and **E**) PIP$_2$ is necessary for Gβγ activation of GIRK2. Currents were recorded from a bilayer containing either GIRK2 alone (**C**) or GIRK2 and 2% brain PIP$_2$ (**E**) before and after application of Gβγ vesicles (indicated with arrow). The amplifier gain was adjusted during the gap in the recording in (**E**). Insets in (**D** and **E**) show spontaneous openings of GIRK2 before application of Gβγ (compare to **B** and **C**). (**F**) Current recorded from a bilayer with GIRK2 and PIP$_2$ before and after application of non-lipidated Gβγ (left side of recording) followed by application of lipidated Gβγ vesicles (right side of recording). Lipidated Gβγ (~15 μM) robustly activated GIRK2 while high concentrations (~116 μM) of non-lipidated Gβγ only slightly increased channel activity.

*Figure 2B,C*. It is thus clear in the bilayer assay that PIP$_2$ or Gβγ alone are insufficient to elicit robust channel activation. Both of these ligands together are necessary. While past studies had reached the conclusion that both ligands are important—that one ligand enhances the affinity of the other (*Huang et al., 1998*; *Yamada et al., 1998*; *Logothetis and Zhang, 1999*)—the complexity of cell membranes obscured the simple conclusion so obvious here that both ligands are necessary to open the channel.

*Figure 2F* addresses the importance of the lipid anchor on the Gβγ subunit. The recording on the left side shows that when mutant Gβγ subunits devoid of the Gγ-linked geranylgeranyl lipid moiety (C68S) were added to solution (to ~116 μM concentration) channels did not open robustly. When Gβγ subunits containing the lipid moiety (~15 μM in lipid vesicles) were subsequently applied to the same membrane, large currents were elicited (*Figure 2F*, right side recording). This experiment shows that the lipid anchor is essential. The likely explanation is, by mediating membrane partitioning, the lipid anchor is required to achieve sufficiently high enough concentrations of Gβγ on the membrane surface to activate GIRK2.

Cellular electrophysiologists have referred to G protein activation of GIRK channels as 'membrane delimited' (*Soejima and Noma, 1984*; *Breitwieser and Szabo, 1985*; *Pfaffinger et al., 1985*; *Logothetis et al., 1987*; *Reuveny et al., 1994*; *Wickman et al., 1994*), meaning Gβγ stays stuck to the membrane when it is released from a GPCR and diffuses to its target. Here, this property of membrane targeting mediated by a lipid anchor is nicely replicated in the bilayer assay. The importance of the lipid anchor is also consistent with the orientation of Gβγ in the crystal structure, which shows

the geranylgeranylated C-terminus of Gγ pointed toward the membrane (*Figure 1A*; *Whorton and MacKinnon, 2013*).

## The sidedness of active channels in planar bilayers

In contrast to the membranes of cells, in which channels are inserted with one orientation, in the planar bilayer system channels and Gβγ subunits insert randomly and thus are oriented in both directions. The experiments in *Figure 3* show, despite the dual orientation of proteins in the bilayer membrane, that short chain C8-PIP$_2$ activates only one orientation of GIRK2 channels—corresponding to those channels with their physiologically intracellular surface pointed toward the chamber to which PIP$_2$ was added. Demonstration of this fact made use of the GIRK channel inhibitor tertiapin-Q (TPNQ), a derivative of the peptide toxin tertiapin isolated from honeybee venom (*Jin and Lu, 1999*). TPNQ inhibits GIRK channels by binding to a specific receptor site on the extracellular surface of the channel. In the first experiment GIRK2 channels and Gβγ subunits were reconstituted into bilayer membranes randomly, C8-PIP$_2$ was added to one side (bottom chamber), and an I-V curve was recorded (*Figure 3A*, squares). The solutions on both sides of the membrane contained equal concentrations of K$^+$ and thus the I-V curve reverses at zero mV, but still exhibits the characteristic inward rectification of the GIRK channel. Next, TPNQ was added to the opposite side (top chamber) and essentially all of the currents were inhibited (*Figure 3A*, circles). The second experiment is similar to the first except C8-PIP$_2$ and then TPNQ were added to the same side of the membrane (top chamber) (*Figure 3C*). In this case TPNQ did not block the K$^+$ currents.

The interpretation of each experiment is shown pictorially (*Figure 3B,D*). C8-PIP$_2$ (and Gβγ) activate a uniformly oriented population of channels from their intracellular surface, whereas oppositely oriented channels are not activated because they are not exposed to PIP$_2$ on their physiologically intracellular side. This conclusion neatly accounts for complete inhibition of currents when TPNQ is added to the side opposite PIP$_2$ (*Figure 3A,B*) and no inhibition when it is added to the same side (*Figure 3C,D*).

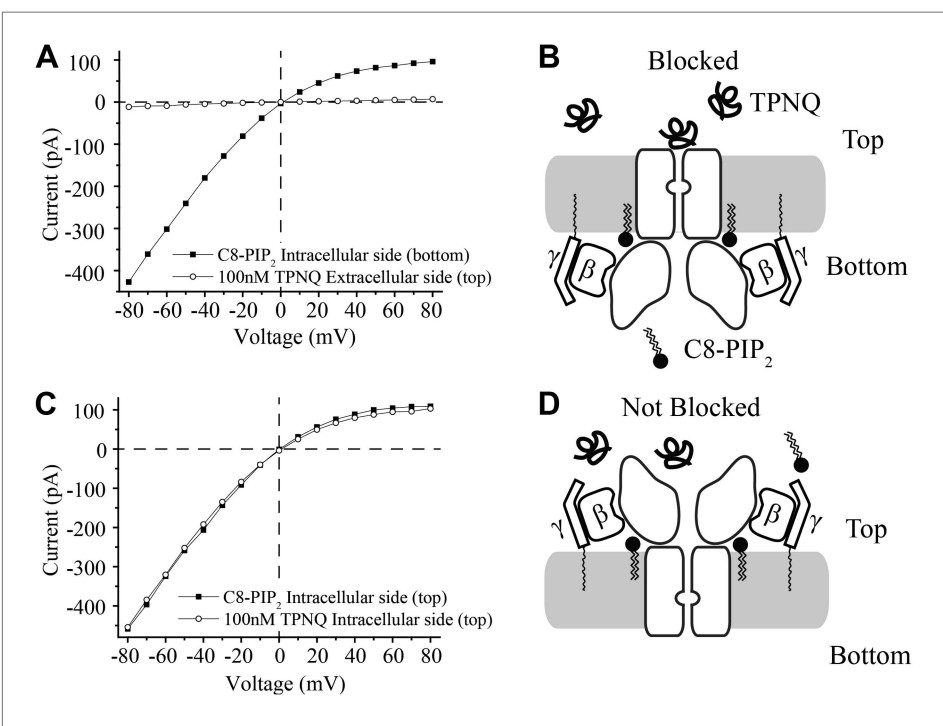

**Figure 3**. Activation of GIRK$_2$ by C8-PIP2 is sided. (**A**) Representative current–voltage relationship recorded from a bilayer containing GIRK2 and Gβγ. Robust current was recorded upon addition of C8-PIP$_2$ to the bottom side of the bilayer (black squares) that is blocked by application of TPNQ to the top side (white circles). (**B**) Schematic of molecular interactions that explain the data in (**A**). (**C** and **D**) Analagous to (**A** and **B**), except that C8-PIP$_2$ and TPNQ were applied to the same side of the bilayer. GIRK2 channels that are activated by C8-PIP$_2$ 'cytoplasmically' can only be blocked by 'extracellular' TPNQ.

## The role of Gα$_{i1}$ in GIRK2 channel regulation

The Gα and Gβγ subunits act on specific target proteins to regulate their function. With GDP bound to the catalytic site of Gα, Gα binds to Gβγ, rendering both (Gα and Gβγ) unavailable to their targets. The role of G protein coupled receptors (GPCRs) is, upon stimulus, to catalyze GDP/GTP exchange on Gα and release of both Gα(GTP) and Gβγ to action. After a period of time, being a slow GTPase, Gα catalyzes the conversion of GTP back to GDP, allowing Gα(GDP) to again form a tight complex with Gβγ, thus terminating action (*Gilman, 1987*).

Gβγ is known to activate GIRK, but as stated in the introduction, several lines of evidence indicate that Gα can also interact directly with the GIRK channel (*Huang et al., 1995*; *Rebois et al., 2006*; *Rubinstein et al., 2007*; *Schreibmayer, 2009*; *Berlin et al., 2010, 2011*). The bilayer assay, in which every component is defined, allows a simple approach to address whether Gα plays a direct role in GIRK channel gating. When Gα$_{i1}$ containing the non-hydrolyzable GTP analog GTPγS is applied to membranes containing GIRK channels there is no activation (*Figure 4A*). In this experiment GIRK2 channels were pre-fused with bilayer membranes containing 1% brain PIP$_2$ (*Balla, 2013*). The characteristic brief baseline channel openings are observed, demonstrating that channels are present in the membrane, but no change in current is observed upon subsequent fusion of vesicles containing Gα$_{i1}$(GTPγS). To confirm the presence of abundant channels, Gβγ-containing vesicles were subsequently fused with the same bilayer membrane and large K$^+$ currents were evoked (*Figure 4A*, right side of trace). From this experiment it is clear that GIRK2 channels are activated by Gβγ but not by Gα$_{i1}$(GTPγS), as summarized pictorially (*Figure 4B*).

While Gα$_{i1}$(GTPγS) fails to affect GIRK2 channel gating in the absence of Gβγ, there remains the possibility that Gα$_{i1}$(GTPγS) can function together with Gβγ to influence gating. To test this possibility we first fused GIRK2 channels and Gβγ into the bilayer and supplemented the top chamber with 32 μM C8-PIP$_2$, followed by fusion of vesicles containing Gα$_{i1}$(GTPγS). C8-PIP$_2$ was used instead of brain PIP$_2$

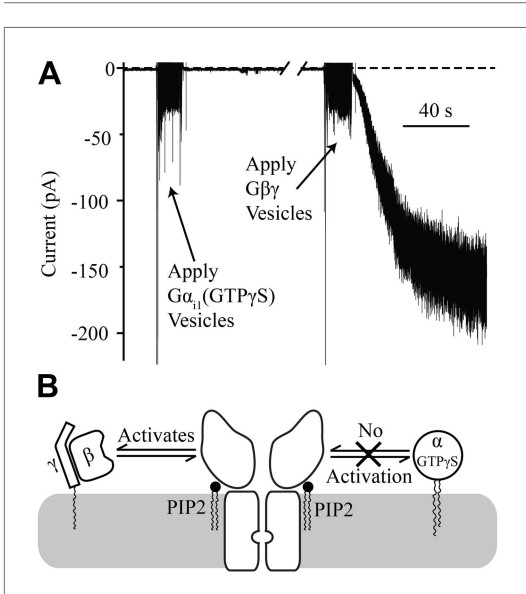

**Figure 4**. Gα$_{i1}$(GTPγS) does not activate GIRK2. Recordings were performed as in *Figure 2*. The increased signal during application of vesicles resulted from opening the chamber for electromagnetic shielding. (**A**) Current recorded from a bilayer with GIRK2 and 1% brain PIP$_2$ before and after application of Gα$_{i1}$(GTPγS) (left) followed by application of Gβγ (right). (**B**) Schematic of molecular interactions that explain the data in (**A**). Activation by Gβγ but not Gα$_{i1}$(GTPγS) confirmed the presence of GIRK2 channels in the bilayer and their insensitivity to applied Gα$_{i1}$.

to ensure that all the activated channels would be oriented with their cytoplasmic side facing the top chamber. To facilitate fusion (*Fisher and Parker, 1984*), vesicles were applied in a solution containing 750 mM KCl. This concentration of KCl near the membrane surface produced a transient reduction of K$^+$ current–until diffusion or mixing restored the local ion composition to bulk values– as shown in the empty vesicle control on the left side of the trace (*Figure 5A*). Fusion of Gα$_{i1}$(GTPγS)-containing vesicles appeared essentially identical to empty vesicles (*Figure 5B*). Thus, GIRK2 channels are unresponsive to Gα$_{i1}$(GTPγS) even after they have been activated by PIP$_2$ and Gβγ.

In contrast, fusion of vesicles containing Gα$_{i1}$(GDP) caused rapid disappearance of activated GIRK2 currents (*Figure 5C*). We also studied the effect of Gα$_{i1}$(GDP) and Gα$_{i1}$(GTPγS) using soluble Gα$_{i1}$ subunits (produced in *Escherichia coli*) that did not contain a covalent membrane lipid anchor. At micromolar concentrations, soluble Gα$_{i1}$(GDP) closed GIRK2 channels, whereas soluble Gα$_{i1}$(GTPγS) had only a small effect most likely due to small amounts of Gα$_{i1}$(GDP) present in the Gα$_{i1}$(GTPγS) preparation (*Figure 5D*). Apparently Gα$_{i1}$(GDP) in solution binds to membrane anchored Gβγ, rendering it unavailable to activate GIRK2, as depicted (*Figure 5E*). It is interesting to contrast the ability of soluble Gα$_{i1}$(GDP) to close GIRK2 channels (*Figure 5D*) with the inability of soluble (lipid anchor-removed) Gβγ to open

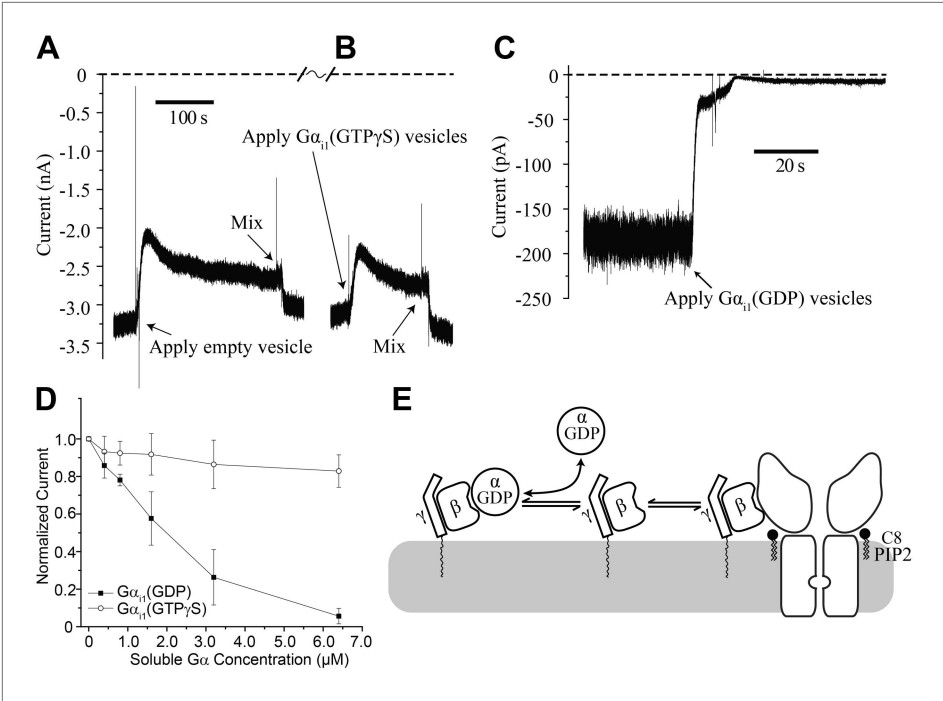

**Figure 5**. Gα$_{i1}$(GDP) but not Gα$_{i1}$(GTPγS) deactivates GIRK2 by sequestering Gβγ. Recordings were performed as in *Figure 2*. (**A** and **B**) Current recording from a bilayer with GIRK2 and Gβγ activated by 32 μM C8-PIP$_2$ before and after (**A**) addition of empty vesicles followed by (**B**) addition of Gα$_{i1}$(GTPγS) vesicles (vesicle additions indicated by arrows). Note that the high salt concentration of the vesicle solution (750 mM KCl) used to facilitate fusion to the bilayer caused a similar transient reduction in K$^+$ current in both (**A**) and (**B**). In both cases, current level returned to near pre-vesicle application levels after thorough mixing. (**C**) Current recording from a bilayer with GIRK2 and Gβγ activated by C8-PIP$_2$ before and after application of vesicles containing Gα$_{i1}$(GDP) (indicated by arrow). (**D**) Comparison of the effects of non-lipidated (soluble) Gα$_{i1}$(GDP) and Gα$_{i1}$(GTPγS) on activated GIRK2. Current at −50 mV holding potential (normalized to the value before addition of Gα$_{i1}$) recorded from bilayers (±SEM, n = 3 bilayers) with GIRK2 and Gβγ activated by C8-PIP$_2$ vs concentration of added Gα$_{i1}$ species is plotted. The lines connecting data points have no theoretical meaning. (**E**) Illustration of molecular interactions that explain data in (**A**–**D**).

GIRK2 channels (*Figure 2F*). This observation suggests differences in the relative affinities of Gβγ for GIRK2 vs Gα$_{i1}$(GDP) (The equilibrium dissociation constant for Gα$_{i1}$(GDP) binding to Gβγ is ~3 nM in Lubrol and ~10 nM in lipids [*Sarvazyan et al., 1998*]). These differential affinities are likely important for G protein regulation of channel activity.

## PIP$_2$ mole fraction dependence of GIRK2 channel activation

The importance of the signaling lipid PIP$_2$ to GIRK channel gating has been well documented through studies in cells (*Huang et al., 1998*; *Sui et al., 1998*; *Rohacs et al., 2002*; *Gamper and Rohacs, 2012*). Previous studies augmented PIP$_2$ levels in cell membranes through the addition of PIP$_2$ to undetermined baseline levels or reduced PIP$_2$ levels through the use of PIP$_2$-sequestering antibodies (*Huang et al., 1998*; *Logothetis and Zhang, 1999*; *Rohacs et al., 2002*; *Inanobe et al., 2010*). These approaches yielded qualitative data on the relationship between GIRK channel activity and PIP$_2$ concentration. Here we aim to describe the quantitative relationship between GIRK channel activity and PIP$_2$ levels.

The bilayer allows quantitative control of lipid composition. In a first experiment GIRK2 channels and Gβγ were reconstituted into planar lipid membranes without PIP$_2$. Soluble C8-PIP$_2$ was then perfused with a microperfusion pipette aimed directly at the membrane to activate channels: during perfusion of C8-PIP$_2$ channels opened and during perfusion of buffer they closed as soluble C8-PIP$_2$ diffused away from the membrane (*Figure 6A*). Repeating this experiment with different C8-PIP$_2$ concentrations, or applying different concentrations of C8-PIP$_2$ to the bath solution, gave the titration data shown (*Figure 6B*). In the graph, in order to compare current levels from different membranes, which in general contain different channel numbers, data were normalized to the current value measured

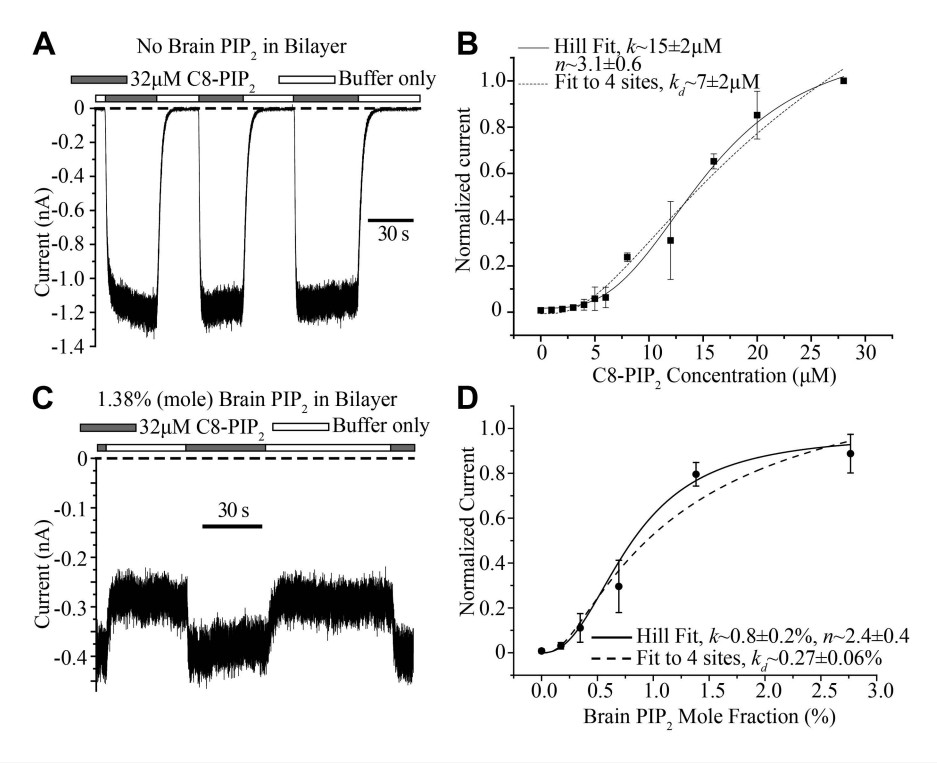

**Figure 6**. Concentration dependence of GIRK2 activation by C8- and brain-PIP$_2$. (**A**) Current recorded from a bilayer containing GIRK2 and Gβγ during local perfusion of C8-PIP$_2$ (black bars above recording) or buffer (white bars above recording). (**B**) Plot of current recorded from bilayers with GIRK2 and Gβγ (normalized to current at 28 μM C8-PIP$_2$, mean ± SEM, n = 3 bilayers) vs concentration of C8-PIP$_2$. Hill fit (solid line) gives an apparent dissociation constant of ~15 μM and a Hill coefficient of ~3.1. Fit to a non-cooperative model ('Materials and methods') in which simultaneous binding of 4 PIP$_2$ molecules to one channel is required for channel opening is also shown (dotted line). (**C**) Current recorded from a bilayer with GIRK2, Gβγ and 1.38% (mole fraction) brain PIP$_2$ during local perfusion of 32 μM C8-PIP$_2$ (black bars above recording) or buffer (white bars above recording). Channels oriented with their extracellular side facing the PIP$_2$ perfusion chamber were blocked with 100 nM TPNQ in perfusion buffers. As a way to normalize channel numbers in different membranes, the current value during perfusion of buffer was normalized to the current level during perfusion of 32 μM C8-PIP$_2$. (**D**) Plot of current recorded from bilayers with GIRK2 and Gβγ (mean ± SEM, n = 3 bilayers) vs concentration of brain PIP$_2$ in the membrane. Regression to Hill equation (solid line) resulted in an apparent dissociation constant of ~0.8% mole fraction brain PIP$_2$ and a Hill coefficient of ~2.4. The dashed line shows regression to the same non-cooperative four sites model as in (**B**).

in the presence of 28 μM C8-PIP$_2$. The activation curve follows a sigmoid shaped concentration dependence, consistent with activation by multiple PIP$_2$ molecules. The atomic structure shows four binding sites per channel (**Figure 1A**). The solid and dashed curves correspond to two different models of channel activation: the solid curve to the Hill equation, which assumes cooperativity between sites, and the dashed curve to a model in which all four binding sites must be occupied but binding is independent (non-cooperative). Both models conform to the data reasonably well. The most important point is that the shape of the titration curve is consistent with multiple binding events being required for channel activation.

C8-PIP$_2$ is a soluble form of PIP$_2$ that partitions into the membrane. Given that the (unknown) partition coefficient is likely independent of C8-PIP$_2$ concentration, the titration in **Figure 6B** is valid for assessing the shape of the titration curve. But what we really want to know is the activity dependence of GIRK2 as a function of the mole fraction of PIP$_2$ inside the membrane. The practical difficulty in determining this is each measurement for a given mole fraction has to come from a different membrane; and each membrane has a different number of channels. In other words, to compare membranes we have to normalize with respect to the number of channels. To do this we produced membranes with a desired mole fraction of full-length PIP$_2$ in the lipid mixture, recorded GIRK2 currents,

and then applied C8-PIP$_2$ to maximally activate (or nearly so), as shown (**Figure 6C**). Data from different bilayers could then simply be normalized with respect to their maximally activated current. The graph shows GIRK2 channel activation as a function of PIP$_2$ mole fraction (**Figure 6D**). Here, as in the case of soluble C8-PIP$_2$, activation is a sigmoid shaped function of PIP$_2$ mole fraction, consistent with a requirement of multiple PIP$_2$ molecules in channel activation. Moreover, the steepest portion of the graph (corresponding to a few tenths to one percent PIP$_2$) is in agreement the known concentration range of PIP$_2$ in the inner leaflet of cell membranes (**Heck et al., 2007**; **van Meer et al., 2008**; **Balla, 2013**). This mole fraction dependence suggests that naturally occurring variations of PIP$_2$ in cell membranes should influence GIRK2 activity level.

## Na$^+$ concentration dependence of GIRK2 channel activation

Intracellular Na$^+$ was shown previously to be an activator of GIRK2 channels (**Petit-Jacques et al., 1999**; **Ho and Murrell-Lagnado, 1999a**, **1999b**; **Inanobe et al., 2013**). Here we aim to replicate Na$^+$ dependence in the bilayer through quantitative measurements and to further our understanding of Na$^+$ activation by investigating the relationship between Na$^+$ and PIP$_2$ activation. **Figure 7** graphs GIRK2 channel activity as a function of internal Na$^+$ or C8-PIP$_2$ concentration (**Figure 7A–C**). In these experiments GIRK2 channels and twofold excess Gβγ (per channel subunit) were reconstituted into bilayer membranes. The 3 dimensional plot shows how these ligands' effects are interdependent (**Figure 7A**). As shown earlier, robust channel opening is absolutely dependent on the presence of both PIP$_2$ and Gβγ (**Figure 7C**). By contrast, GIRK2 channels open in the absence of internal Na$^+$, but opening increases as the Na$^+$ concentration is increased. Viewed along the axis of PIP$_2$ activation, the presence of internal Na$^+$ steepens the dependence of channel opening on PIP$_2$ concentration (**Figure 7B**). Thus, Na$^+$ is not essential but augments channel opening. The effect of Na$^+$ depends on the degree of activation by PIP$_2$. In the presence of ample PIP$_2$, channel activity increases as a function of Na$^+$ concentration most steeply in the range 0–10 mM. This dependence seems well matched to the concentration range over which cytoplasmic Na$^+$ levels change in response to excessive electrical excitation (**Muller and Somjen, 2000**; **Somjen, 2002**).

## Discussion

One question addressed in this study, motivated by past work showing that Gα can interact physically with at least some GIRK channels (**Yatani et al., 1988**; **Huang et al., 1995**; **Peleg et al., 2002**; **Ivanina et al., 2004**; **Berlin et al., 2011**), concerns the role of Gα vs Gβγ in GIRK channel activation. Using

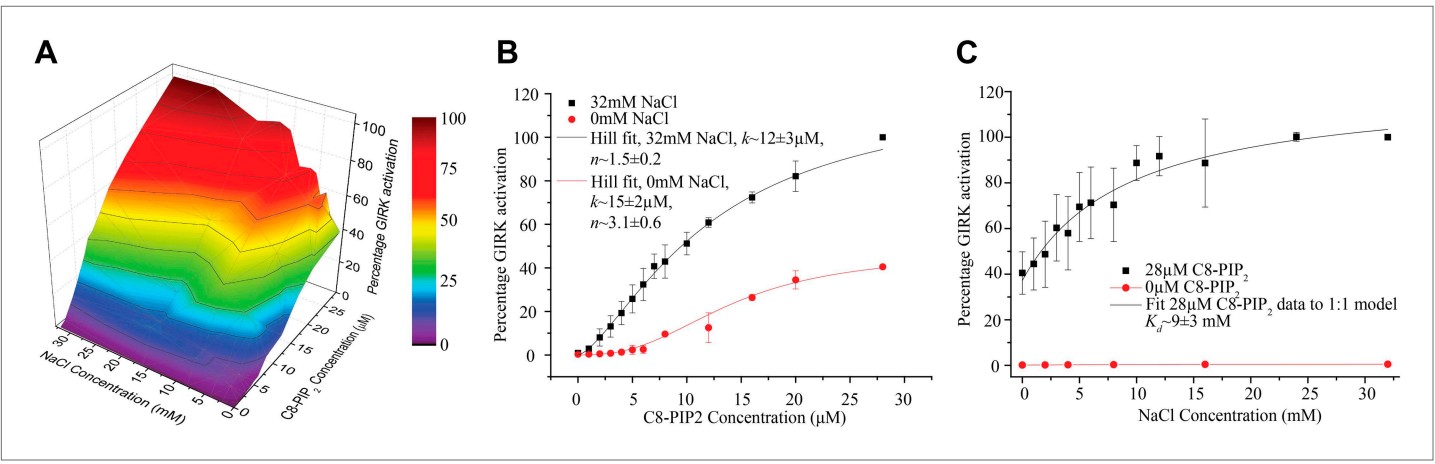

**Figure 7**. Interdependence of PIP$_2$ and Na$^+$ activation. Channel activity is normalized to that at the condition in which 28 μM of PIP$_2$ and 32 mM of Na$^+$ is present. (**A**) A 3-D representation of the activation landscape of PIP$_2$ and Na$^+$. The surface is gradient colored according to activation. (**B**) Plot of percentage activity of GIRK2 channels in membranes with saturating Gβγ vs added C8-PIP$_2$ concentration in the absence (red, n = 3) and presence (black, n = 3) of 32 mM Na$^+$. PIP$_2$ activated GIRK2 currents were ~2.5-fold higher (at 28 μM) in the presence of 32 mM Na$^+$. Hill fitting resulted in apparent dissociation constants of ~15 μM and ~12 μM and Hill coefficients of ~3.1 and ~1.5 in the absence and presence of Na$^+$, respectively. (**C**) Plot of percentage activity of GIRK2 channels in membranes with saturating Gβγ vs concentration of added Na$^+$ in the absence (red, n = 3) and presence (black, n = 3) of 28 μM C8-PIP$_2$. The titration data were fit to a simple 1:1 dose response model with an apparent dissociation constant of ~9 mM.

bilayer reconstitution we find the following. Gβγ activates GIRK2, as was previously shown (*Logothetis et al., 1987*; *Reuveny et al., 1994*; *Wickman et al., 1994*; *Kofuji et al., 1995*). $Gα_{i1}$(GTP) does not activate GIRK2 and moreover does not appear to influence the activation of GIRK2 by Gβγ. These findings suggest, at least for the case of the activated form of $Gα_{i1}$ ($Gα_{i1}$(GTP)) and GIRK2, that physical interactions between these proteins, if they indeed exist, are without direct functional consequence.

$Gα_{i1}$ inhibits the GIRK2 channel activity only when it is in its GDP bound form, consistent with the canonical view that GIRK is activated by Gβγ and that GDP-bound $Gα_{i1}$ sequesters Gβγ (*Gilman, 1987*; *Wickman et al., 1994*; *Peleg et al., 2002*). In the bilayer experiments the action of $Gα_{i1}$(GDP) is so rapid (*Figure 5C*) as to suggest that it might bind to the channel and cause Gβγ to dissociate. We think, however, that such a displacement mechanism need not be invoked to account for the rapid inhibitory effect of $Gα_{i1}$(GDP). In drawing this conclusion we appeal to the inefficiency of Gβγ to activate when it does not contain a lipid anchor. Presumably Gβγ fails under this circumstance because, without membrane partitioning, its local concentration near the channel is never high enough to achieve sufficient occupancy to open the channel. Thus, Gβγ apparently binds to the channel with low affinity, compatible with a rapid dissociation rate. Comparatively, Gβγ appears to bind $Gα_{i1}$(GDP) with high affinity, as this would account for sequestration of Gβγ even when no lipid is present on $Gα_{i1}$(GDP) (*Figure 5D*). Therefore, it seems likely that $Gα_{i1}$(GDP) turns off GIRK by simply binding to the free (channel unbound) form of Gβγ and shifting the equilibrium away from channel occupancy by Gβγ.

Several further lines of evidence support the notion that Gβγ binds with low affinity to the GIRK channel. During biochemical purification we have never detected a complex between these two proteins on gel filtration even when both proteins are at concentrations in the 100 μM range (*Whorton and MacKinnon, 2013*). Using solution NMR spectroscopy, Shimada et al. estimated the affinity between Gβγ and the GIRK1 cytoplasmic domain (which accounts for the full protein–protein interaction surface) to be about 250 μM (*Yokogawa et al., 2011*). Using this value, if we conservatively assume an association rate of $10^5$ $M^{-1}s^{-1}$ (well below the expected diffusion limit in solution) then the dwell time for a single Gβγ on the channel is calculated to be about 100 ms, a duration much shorter than the wash out time in our experiments. Finally, a low affinity interaction matches the relatively small surface area (~700 $Å^2$) of interaction between Gβγ and its binding site on GIRK2 in the crystal structure (*Whorton and MacKinnon, 2013*). Therefore, all observations are consistent with the idea that Gβγ is in rapid exchange with its binding site on the GIRK channel.

Given the apparent low affinity between Gβγ and the GIRK channel, the high affinity of Gβγ for Gα(GDP), and the absence of an effect of Gα on channel function, we think the following mechanism is most likely. GPCR activation generates Gβγ and Gα(GTP), then Gβγ activates GIRK, but once GTP is hydrolyzed the resultant Gα(GDP) removes Gβγ from the channel by equilibrium-mass action. This picture is simple and in good agreement with the accepted view, but it raises the following question. If the affinity of Gβγ for GIRK is so low and the interaction so transient, how does GPCR activation ever achieve sufficiently high concentrations of Gβγ in the membrane to activate the channel? It seems unlikely that a single GPCR located near a GIRK channel would be able to generate a high enough local concentration of subunits to activate it, especially if multiple bound Gβγ subunits are required, which we suspect is the case on the basis of the crystal structure (*Whorton and MacKinnon, 2013*). One way to explain this dilemma is to imagine that GIRK channels and GPCRs might be co-localized in relatively large patches on the membrane. Such an organization would exploit a fundamental property of 2-dimensional diffusion: a point source in an infinite space has no steady state solution. In other words, the transition between local and bulk Gβγ concentration will grow radially with activation time. Thus, in a patch of multiple GPCRs (and channels) a sufficiently high concentration of Gβγ on the membrane could be reached. The possibility of co-localization of GPCRs and GIRK channels raises an interesting challenge to the replication of cell-like signaling in the bilayer. It could be that additional cellular elements (e.g., cytoskeleton) are required to perfectly replicate the signaling behavior observed in cells.

Another question addressed in this study concerns the quantitative dependence of GIRK2 channel activity on the concentration of $PIP_2$ in the membrane. In the presence of Gβγ, GIRK2 channel activity shows a sigmoid dependence on $PIP_2$ mole fraction. A Hill plot yields a coefficient around 2.5. We cannot tell from our data whether $PIP_2$ binding is cooperative (as the Hill model assumes) or non-cooperative (e.g., if multiple $PIP_2$ molecules must bind, but independently, to open the channel).

Cooperative or not, the sigmoid functional relationship points to multiple $PIP_2$ molecules required to open the channel. This result is consistent with the crystal structures, which show four $PIP_2$ molecules bound to the channel (*Whorton and MacKinnon, 2011*). The apparent binding constant for $PIP_2$ is 0.8% mole fraction (Hill model), which, together with the Hill coefficient, describes the position of the steepest part of the sigmoid curve along the mole fraction axis. We see that GIRK2 activity is most responsive to $PIP_2$ in the mole fraction range 0.1–1.0%. This is precisely the range over which $PIP_2$ is known to vary in the inner membrane leaflet of cells (*Balla, 2013*). In other words, GIRK2 is poised to be under control by $PIP_2$ membrane concentration.

Finally this study addressed the interdependence of multiple ligands known to activate GIRK channels. Robust channel opening requires the presence of both Gβγ and $PIP_2$. Earlier electrophysiological studies concluded that activation can occur at high concentrations of a single ligand and that the role of Gβγ, for example, was to enhance the stimulation brought about by $PIP_2$ (*Huang et al., 1998*; *Sui et al., 1998*). Those studies were carried out in cells, where the composition of the membrane is not so certain. The planar bilayer experiments support a simple conclusion: Gβγ and $PIP_2$ are simultaneously required to activate GIRK2. This conclusion is beautifully compatible with structural studies of GIRK2, which show that when $PIP_2$ alone is bound the channel's inner gate remains closed (it exhibits the same closed conformation as when $PIP_2$ is not bound) (*Whorton and MacKinnon, 2011*). But when both Gβγ and $PIP_2$ are bound the cytoplasmic domain rotates with respect to the transmembrane domain and causes the inner helix gate to begin opening (*Whorton and MacKinnon, 2013*). $PIP_2$ binds near the interface between the cytoplasmic domain and the transmembrane domain. We hypothesize that $PIP_2$ strengthens allosteric communication between these two domains of the channel. Such a strengthening of interaction would allow conformational changes in the cytoplasmic domain, induced by Gβγ binding, to open the gate in the transmembrane domain.

$Na^+$ is not sufficient and not necessary for GIRK2 channel activation. $Na^+$ increases channel opening in the presence of Gβγ and $PIP_2$ and thus $Na^+$ is best described as a modulator of the other ligands' effects (*Figure 7C*). This conclusion is easily appreciated through inspection of the $C8$-$PIP_2$ activation curve in the presence and absence of $Na^+$ (*Figure 7B*). Not only does $Na^+$ increase the current level at any given concentration of $PIP_2$ (and a fixed concentration of Gβγ), but the shape of the $PIP_2$ dependence curve is affected by $Na^+$. In other words, $Na^+$ does not simply scale the $PIP_2$ dependence curve. This is explicable if the binding affinity/effect of $PIP_2$ is coupled to the presence of $Na^+$ (*Ho and Murrell-Lagnado, 1999a*, *1999b*). In the crystal structure $Na^+$ binds at a location in between Gβγ and $PIP_2$. It is therefore reasonable to think that conformational changes induced by the binding of one ligand, for example $Na^+$, would have consequences for the binding/effect of another ligand, for example Gβγ or $PIP_2$. Although $Na^+$ plays a modulatory rather than obligatory role in gating, its effect is substantial and occurs over the range in which cytoplasmic $Na^+$ concentration is known to change when a cell is experiencing excessive electrical excitation (*Muller and Somjen, 2000*). These findings are in good agreement with the hypothesis that $Na^+$ modulation renders GIRK2 a negative feedback safety element (*Ho and Murrell-Lagnado, 1999a*).

The dual requirement for both Gβγ and $PIP_2$ may have important consequences for the control of cellular electrical excitability by different signaling pathways. Consider that Gβγ is released upon stimulation of all GPCRs, but GIRK channels only open specifically in response to $G_{i/o}$ coupled GPCRs ($G_{i/o}$ refers to G protein trimers containing i/o class Gα subunits) (*Breitwieser and Szabo, 1985*; *Pfaffinger et al., 1985*). It is easy to understand a possible origin of specificity when comparing $G_{i/o}$ coupled GPCRs (which activate GIRK) and $G_q$ coupled GPCRs (which do not activate GIRK) (*Lei et al., 2001*; *Cho et al., 2005*). $G_q$ coupled GPCRs release Gβγ, but at the same time the $Gα_q$ subunit activates PLC, which depletes $PIP_2$ through hydrolysis (*Cho et al., 2005*; *Keselman et al., 2007*; *Sohn et al., 2007*). In the absence of sufficient $PIP_2$, Gβγ cannot activate GIRK. Indeed, recent experiments have shown in both atrial pacemaker cells (*Cho et al., 2005*) and certain neurons (*Sohn et al., 2007*; *Yamamoto et al., 2014*) that $G_q$ coupled GPCR stimulation inhibits GIRK activation by $G_{i/o}$ coupled GPCR stimulation. It is thus reasonable to think that absence of $PIP_2$ accounts at least in part for the inability of $G_q$ coupled GPCRs to activate GIRK. $G_{i/o}$ coupled GPCRs on the other hand generate Gβγ without degrading $PIP_2$ so both necessary ligands are available to activate GIRK. This kind of GPCR selectivity results because GIRK is an AND gate with respect to Gβγ and $PIP_2$ activation. In addition to the more widespread (but not yet proved) idea of GPCR/target protein co-localization (*Fowler et al., 2007*; *Labouebe et al., 2007*; *Cui et al., 2010*; *Luscher and Slesinger, 2010*), AND gating of GPCR targets by multiple ligands provides a mechanism for selectivity in G protein signaling.

## Materials and methods

### Protein expression and purification

Mouse GIRK2 (amino acid residues 52–380) was expressed in *Pichia pastoris* cells, extracted with decyl-β-D-maltopyranoside (DM) and purified as previously described (*Whorton and MacKinnon, 2011*). The protein was concentrated to ~15 mg/ml after purification and used immediately for reconstitution.

Human G protein subunits $β_1$ and $γ_2$ were expressed in High Five (Invitrogen, Carlsbad, CA) insect cells by co-infection as previously described (*Whorton and MacKinnon, 2013*). Cell membranes were then prepared and G proteins were extracted from the membranes with Na-cholate (Sigma, St. Louis, MO) and purified as described (*Whorton and MacKinnon, 2013*) with the following modifications: 1% (wt/vol) of Na-Cholate was used in Talon resin (Clontech, Mountain View, CA) purification instead of 0.5% (wt/vol) of Anzergent 3–12 (Anatrace, Maumee, OH). After elution from Talon resin and digestion with PreScission protease, the protein was diluted to an imidazole concentration of 10 mM. The diluted solution was passed through a column packed with Talon resin pre-equilibrated with the same buffer without imidazole. The flow-through from this step was concentrated and loaded onto a Superdex 200 10/300 (GE Healthcare, Pittsburgh, PA) gel filtration column in the buffer 20 mM HEPES pH 8.0, 150 mM KCl, 5 mM DTT, 1 mM EDTA, 0.2% DM. The hetero-dimer Gβγ protein eluted as a major peak at ~12 ml. Fractions containing this peak were pooled and concentrated to ~30 mg/ml and were either used immediately or frozen in aliquots at −80°C.

The non-lipidated form of Gβγ was expressed by co-infection of High Five cells with two baculoviruses each containing $Gβ_1$ and $Gγ_2$ C68S mutant genes. The purification methods are the same as wild-type Gβγ except that no detergent was used.

For the preparation of $Gα_{i1}$ protein, cDNA of human $Gα_{i1}$ was cloned into pFastBac (Invitrogen) vector without any affinity purification tag. High Five insect cells were co-infected with three baculoviruses each bearing one of the following proteins: $Gα_{i1}$, $Gβ_1$ and $Gγ_2$. $Gα_{i1}$ was purified by first binding the $Gα_{i1}βγ$ heterotrimer to Talon resin and then eluting just the $Gα_{i1}$ subunit by dissociating it from the heterotrimer with aluminum tetrafluoride $AlF_4^-$ as previously described (*Kozasa, 2004*). Briefly, cell membranes containing all three G protein subunits were extracted with Na-cholate in the buffer system used for Gβγ purification supplemented with 3 mM $MgCl_2$ and 10 µM GDP. After binding the extracted protein to Talon resin and washing with 10 mM imidazole, $Gα_{i1}$ was eluted in the same buffer supplemented with 50 mM $MgCl_2$, 30 µM $AlCl_3$, 10 mM NaF, and 30 µM GDP. Since this elution step specifically disrupts the $Gα_{i1}$-Gβγ interaction, the eluted fraction is fairly pure. The eluent is then concentrated and loaded onto a Superdex 200 10/300 gel filtration column in the buffer 20 mM HEPES pH 8.0, 150 mM KCl, 5 mM DTT, 2 mM MgCl2, 0.2% DM. $Gα_{i1}$ eluted as a major peak at ~14.5 ml. Fractions containing this peak were pooled and concentrated to ~10 mg/ml. For exchange of nucleotide on $Gα_{i1}$, either 2 mM GDP or GTPγS were added to the protein, then incubated at 37°C for 30 min. Slight precipitation usually occurs, but can be clarified by centrifugation at 4°C. The binding status of $Gα_{i1}$ was assayed by a controlled trypsin digestion assay (*Mazzoni et al., 1991*; *Marin et al., 2001*) in which the activated form of $Gα_{i1}$ (GTPγS) is protected from digestion by N-tosyl-L-phenylalanyl chloromethyl ketone (TPCK) treated trypsin. Purified $Gα_{i1}$ proteins were put on ice until used.

To obtain non-lipidated $Gα_{i1}$ protein, $Gα_{i1}$ cDNA was cloned into pET-28a(+) vector (Novagen, Billerica, MA) with an N-terminal hexahistidine tag and transformed into BL21 *E. coli* for protein expression. In order to obtain non-lipidated protein, we did not co-express with N-myristoyltransferase (NMT) (*Mumby and Linder, 1994*). At an $OD_{600}$ of ~0.9. 1 mM IPTG was used to induce expression, which was continued at 37°C for 8 hr. The same buffers were used for non-lipidated $Gα_{i1}$ purification as for non-lipidated Gβγ except that $MgCl_2$ was added to 2 mM and GDP to 10 µM. The purification steps are essentially the same as non-lipidated Gβγ. After gel filtration, protein was concentrated and nucleotide exchange (GDP or GTPγS) was performed as described above for lipidated $Gα_{i1}$. The function of the proteins were tested with a histidine tag based pull down assay using Gβγ.

### Reconstitution of proteoliposomes

A lipid mixture composed of 3:1 (wt:wt) 1-palmitoyl-2-oleoyl-sn-glycero-3-phosphoethanolamine (POPE) : 1-palmitoyl-2-oleoyl-sn-glycero-3-phospho-(1′-rac-glycerol) (POPG) was used for reconstitution of GIRK2 channels and G proteins into lipid vesicles. The reconstitution procedure is similar to that previously reported (*Heginbotham et al., 1999*; *Long et al., 2007*; *Tao and MacKinnon, 2008*;

*Brohawn et al., 2012*) with some modifications. Briefly, 20 mg/ml of the above lipid mixture was dispersed by sonication and then solubilized with 20 mM DM. GIRK2 and Gβγ were diluted with reconstitution buffer (10 mM potassium phosphate pH 7.4, 150 mM KCl, 1 mM EDTA and 3 mM DTT) supplemented with 0.2% DM to 2 mg/ml (~51 μM) and 4.8 mg/ml (~106 μM), respectively. Equal volumes of 20 mg/ml solubilized lipid mixture was combined with each of these protein solutions to make protein: lipid (wt:wt) ratios of 1:10 and 1:4.2 with a final lipid concentration of 10 mg/ml. For co-reconstitution of GIRK2 with Gβγ, the two proteins were mixed, resulting in final concentrations of 2 mg/ml and 4.8 mg/ml respectively. Subsequently, this solution was mixed with an equal volume of 20 mg/ml solubilized lipid mixture. Detergent was removed by dialysis against reconstitution buffer at 4°C for 4–6 days. For reconstitution of Gα$_{i1}$(GTPγS) and Gα$_{i1}$(GDP), the proteins were diluted to 4 mg/ml (~100 μM) with reconstitution buffer supplemented with 0.2% DM and 2 mM of respective nucleotides. After mixing the diluted Gα$_{i1}$ protein solutions with equal volumes of 20 mg/ml solubilized lipid mixture, the detergent was removed by dialysis against reconstitution buffer supplemented with either 10 μM GTPγS or GDP, at 4°C for 4–6 days. The resulting proteoliposomes were flash-frozen with liquid nitrogen in 20 μl aliquots and stored at −80°C until needed.

## Electrophysiology

Bilayer experiments were performed as previously described (*Ruta et al., 2003*) with the following modifications: 20 mg/ml of a lipid solution in decane composed of 2:1:1 (wt:wt:wt) of 1,2-dioleoyl-*sn*-glycero-3-phosphoethanolamine (DOPE) : 1-palmitoyl-2-oleoyl-*sn*-glycero-3-phosphocholine (POPC) : 1-palmitoyl-2-oleoyl-*sn*-glycero-3-phospho-L-serine (POPS) was painted over a ~100 μm hole on a piece of polyethylene terephthalate transparency film that separates two chambers in a polyoxymethylene block (*Figure 1B*). In some experiments, 1–4% (wt%) of brain PIP$_2$ (Avanti Polar Lipids, Alabaster, AL) was added into the lipid mixture. In most experiments the same buffer (10 mM potassium phosphate pH 7.4, 150 mM KCl, 1 mM EDTA) was used in both chambers. Voltage across the lipid bilayer was clamped with an Axopatch 200B amplifier in whole-cell mode. The analog current signal was low-pass filtered at 1 kHz (Bessel) and digitized at 20 kHz with a Digidata 1322A digitizer. Digitized data were recorded on a computer using the software pClamp (Molecular Devices, Sunnyvale, CA). For local perfusion, a SmartSquirt8 ValveLink Micro-Perfusion System (Automate Scientific, Berkeley, CA) was used with a 250 μm diameter perfusion pencil positioned just above the lipid bilayer. Recordings were performed at room temperature.

The Hill equation used for fitting titration data is:

$$I = I_0 + (I_{\max} - I_0)\frac{x^n}{k^n + x^n}, \tag{1}$$

where $I$ is the observed current, $I_0$ is the basal current (spontaneous opening current and leak, very close to 0), $I_{\max}$ is the maximum current when channels are fully activated, $x$ is ligand concentration and $n$ is the Hill coefficient.

Under the assumption that binding of each PIP$_2$ to GIRK2 is non-cooperative (independent), the fraction of PIP$_2$ binding sites occupied was expressed as:

$$q = \frac{x}{k_d + x}, \tag{2}$$

where $q$ is the fraction of occupied PIP$_2$ binding sites, $x$ is ligand concentration and $k_d$ is the apparent dissociation constant. For a channel with four PIP$_2$ binding sites the fraction of channels with all four sites occupied is:

$$p = \binom{4}{4} \times q^4 \times (1 - q)^0 = q^4. \tag{3}$$

If channel opening corresponds (is equivalent) to occupancy by four PIP$_2$ molecules then the total current is:

$$I = I_{max} \times p, \tag{4}$$

where $I_{\max}$ is the maximum current when all channels are activated. Substituting *Equation 2* and *3* into *Equation 4* gives

$$I = I_{max} \times \left( \frac{x}{k_d + x} \right)^4 . \tag{5}$$

We used *Equation 5* in fitting titration data points. Although likely overly-simplified, the purpose of using this model is to emphasize that a sigmoidal dose–response does not necessarily indicate cooperativity among binding sites.

## Acknowledgements

We thank Joel Butterwick, Xiao Tao and Emily Brown for advice on biochemistry and reconstitution of the channels; Stephen Brohawn for comments on the manuscript; Y Hsiung for assistance with insect cell culture; and members of the MacKinnon laboratory for helpful discussions. This work was supported in part by NIHGM43949. RM is an investigator in the Howard Hughes Medical Institute.

## Additional information

### Funding

| Funder | Grant reference number | Author |
| --- | --- | --- |
| Howard Hughes Medical Institute | | Roderick MacKinnon |
| National Institute of General Medical Sciences | NIHGM43949 | Roderick MacKinnon |

The funders had no role in study design, data collection and interpretation, or the decision to submit the work for publication.

### Author contributions

WW, Conception and design, Acquisition of data, Analysis and interpretation of data, Drafting or revising the article; MRW, Conception and design, Drafting or revising the article; RMK, Conception and design, Analysis and interpretation of data, Drafting or revising the article

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
