## [Decision Letter]

Thank you for sending your work entitled “Quantitative analysis of mammalian GIRK2 channel regulation by G proteins, PIP_2_ and Na^+^ in a reconstituted system” for consideration at *eLife*. Your article has been favorably evaluated by John Kuriyan (Senior editor) and Richard Aldrich (Board of Reviewing Editors) and two external reviewers (Donald Hilgemann and Christopher Miller).

The Reviewing editor and the other reviewers discussed their comments before we reached this decision, and the Reviewing editor has assembled the following comments to help you prepare a revised submission.

In this elegant, thorough, and important fundamental analysis of GIRK activation mechanisms, the authors exploit the membrane reconstitution approach for precisely the purposes le bon Dieu (Ef Racker) invented it. For what seems like millennia of time and kilograms of publications on cellular patch-recording, GIRK channels have been understood to be activated on the intracellular side by G-proteins, PIP_2_, and in some case Na^+^ ions. But the complexity of the cell membrane, even when detached from the cell itself, has stymied a quantitative analysis of these effectors. This is not a persnickety pedant's problem of being unable to dot some inconsequential i's and cross some trivial t's, but is a basic deficit that has persistently prevented the field from developing a satisfactory picture of GIRK function. For many channels, such functional landscapes were well worked out long before crystal structures appeared, but for GIRK it's the reverse situation. The various GIRK structures that have been emerging from MacKinnon's lab over past years have been a bit thin on the function that the structures would be so valuable in explaining. Instead, features such as PIP_2_ and Na^+^ activation had almost to be “read into” the new structures rather than being explained by them.

With this paper, the situation changes. Now, the important GIRK effectors are examined with full reductionist control, of both protein and lipid bilayer composition: G-protein subunits charged with relevant nucleotides, PIP_2_ etc, all in quantitatively manipulable amounts. The conclusions are not surprising – the basic lessons have been long suspected, both from qualitative probing by cellular electrophysiology and by MacKinnon's previous GIRK structures – but they are deeply satisfying and now quantitative. In essence, the functions explain the structures, for a refreshing change. Particularly important is – at last – a proper activation curve for PIP_2_, real PIP_2_ from real brains, along with a comparison to the convenient, commonly used exogenous C8-analog. It's also satisfying to see a Hill-fit (nH=2.7) accompanied by an unapologetic assertion that this, while consistent with involvement of 4 binding sites, long-envisioned in cartoons and now seen in crystal structures, cannot adequately distinguish cooperative from independent activator binding. In addition, the effect of Na^+^ as a modulator of PIP_2_ action is clearly shown by direct experiments, simple on paper but never before properly implemented. The authors even treat old-timers with a brief historical frisson by re-visiting the non-role of Gα, lubrol and all, reminding them of the Great Clapham-Brown Debates of yore that established for the first time the central place of Gβγ in GIRK function. Finally, the exposition relates the mechanistic minutiae uncovered here to central biological roles of GIRKs in cardiac and neuronal electrical behaviors.

The main issue that needs to be addressed is that the authors need to more clearly lay out the state of understanding of this system previous to their present contributions. They should more directly describe the context, within the existing knowledge, of each of their experimental conclusions and more greatly emphasize which of their conclusions are confirmatory (but done in a superior experimental system) and which are unprecedented.

---

## [Author Response]

*The authors need to more clearly lay out the state of understanding of this system previous to their present contributions. They should more directly describe the context, within the existing knowledge, of each of their experimental conclusions and more greatly emphasize which of their conclusions are confirmatory (but done in a superior experimental system) and which are unprecedented*.

To this issue the Introduction is (and was) structured with a second paragraph describing what is known: “As background to the questions addressed in the present study, the following conclusions have been established over the past several decades of study…” This is followed by a third paragraph: “The present study addresses the following still unanswered questions…”

For further clarification we have added to the Results section a statement emphasizing the new understanding of an obligatory dual requirement for both Gbg and PIP_2_ and its distinction from past descriptions. This is again presented in the Discussion section on the concept of GIRK as an AND gate.

We have also added to the Results section an introductory paragraph under the subheading “PIP_2_ mole fraction dependence…” describing what was known and what we are aiming to understand. A similar explanatory introduction has been added under the subheading “Na+ concentration dependence…”

The first paragraph of the Discussion (and beginning of the second) has been rewritten to emphasize what was known about Gbg versus Ga interactions with GIRK and why the question of a potential role for Ga is still relevant today. Our experiments show no functional effect, but the case remains unsettled for GIRK1 and Gai3 and other pairings.

We hope these additions are clarifying and not too redundant regards what is confirmatory (i.e., that Gbg activates and Ga does not), what represents a more quantitative understanding (i.e. GIRK2 activation as a function of PIP2 mole fraction and as a function of Na^+^), and what is new (i.e., the absolute requirement for both Gbg and PIP_2_ and the relationship between PIP_2_ and Na^+^ in GIRK2 activation).